# $\mathbb{E}^{\mathbf{FWI}}$: Multiparameter Benchmark Datasets for Elastic Full Waveform Inversion of Geophysical Properties

**Shihang Feng**[1,*]     **Hanchen Wang**[1,*]     **Chengyuan Deng**[1,2]     **Yinan Feng**[1]

**Yanhua Liu**[1,3]     **Min Zhu**[1,4]     **Peng Jin**[1,5]     **Yinpeng Chen**[6]     **Youzuo Lin**[1,7]

[1]Los Alamos National Laboratory  [2]Rutgers University  [3]Colorado School of Mines
[4]University of Pennsylvania  [5]The Pennsylvania State University  [6]Microsoft
[7]University of North Carolina at Chapel Hill
shihang.feng@live.com
{hanchen.wang, charles.deng, ynf, yanhualiu, minzhu, pjin, ylin}@lanl.gov
yiche@microsoft.com, yzlin@unc.edu

## Abstract

Elastic geophysical properties (such as P- and S-wave velocities) are of great importance to various subsurface applications like $CO_2$ sequestration and energy exploration (e.g., hydrogen and geothermal). Elastic full waveform inversion (FWI) is widely applied for characterizing reservoir properties. In this paper, we introduce $\mathbb{E}^{\mathbf{FWI}}$, a comprehensive benchmark dataset that is specifically designed for elastic FWI. $\mathbb{E}^{\mathbf{FWI}}$ encompasses 8 distinct datasets that cover diverse subsurface geologic structures (flat, curve, faults, etc). The benchmark results produced by three different deep learning methods are provided. In contrast to our previously presented dataset (pressure recordings) for acoustic FWI (referred to as OPENFWI), the seismic dataset in $\mathbb{E}^{\mathbf{FWI}}$ has both vertical and horizontal components. Moreover, the velocity maps in $\mathbb{E}^{\mathbf{FWI}}$ incorporate both P- and S-wave velocities. While the multicomponent data and the added S-wave velocity make the data more realistic, more challenges are introduced regarding the convergence and computational cost of the inversion. We conduct comprehensive numerical experiments to explore the relationship between P-wave and S-wave velocities in seismic data. The relation between P- and S-wave velocities provides crucial insights into the subsurface properties such as lithology, porosity, fluid content, etc. We anticipate that $\mathbb{E}^{\mathbf{FWI}}$ will facilitate future research on multiparameter inversions and stimulate endeavors in several critical research topics of carbon-zero and new energy exploration. All datasets, codes[1] and relevant information can be accessed through our website at `https://efwi-lanl.github.io/`.

## 1 Introduction

Seismic imaging is similar to how submarines use sonar to map out underwater landscapes and locate objects. Just as sonar sends out sound waves and interprets the echoes to figure out distances and

---

[*]Equal contribution

[1]Codes will be released upon approval by Los Alamos National Laboratory and U.S. Department of Energy.

37th Conference on Neural Information Processing Systems (NeurIPS 2023) Track on Datasets and Benchmarks.

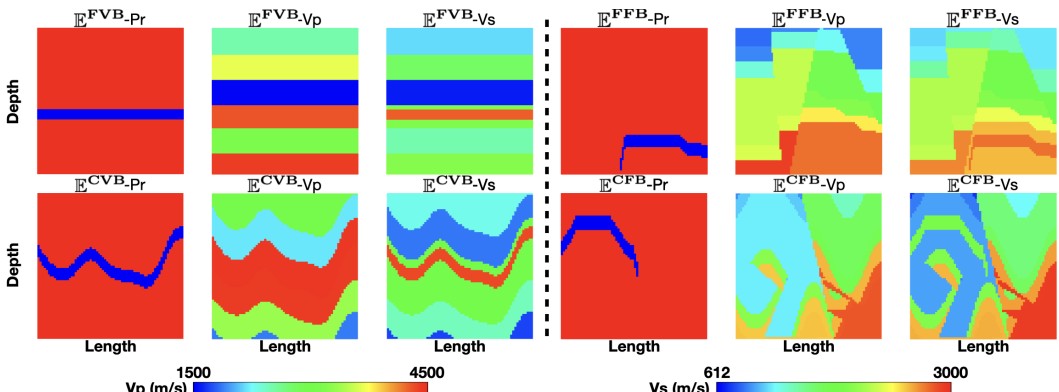

Figure 1: **Gallery of $\mathbb{E}^{\mathbf{FWI}}$**: one example of reservoir structure (Pr) and velocity maps ($V_P$, $V_S$) from the $\mathbb{E}^{\mathbf{FVB}}, \mathbb{E}^{\mathbf{FFB}}, \mathbb{E}^{\mathbf{CVB}}, \mathbb{E}^{\mathbf{CFB}}$ datasets. Pr refers to the designed reservoir, which is the Poisson's ratio anomaly calculated explicitly by $V_P$ and $V_S$, two kinds of wave traveling speed at each spatial point.

shapes of underwater objects, geoscientists send seismic waves deep into the Earth. By analyzing how these waves are reflected back, they can generate detailed images of the subsurface and deduce properties of rock formations.

Seismic waves, propagating through the subsurface medium, can unveil the physical properties of the rock formations. Full waveform inversion (FWI) has emerged as an effective technique for obtaining high-resolution models of the subsurface physical properties [1, 2, 3]. In essence, it's like refining our underwater sonar map to capture more details and nuances. The determination of such properties from seismic data is posed as an inverse problem. This means we use the reflected waves (akin to sonar echoes) to infer the properties of the rocks they passed through. FWI refines this process, striving to find the most accurate representation by minimizing the difference between observed and synthetic seismic data [4]. This technique has made substantial contributions across a range of domains, including geothermal energy exploration, earthquake monitoring, subsurface imaging for engineering applications, and many others [5].

Acoustic approximations have been widely employed in wavefield simulation for FWI, resulting in a substantial reduction in computational cost [6, 7]. It assumes that the subsurface medium behaves as a fluid and focuses on simulating the kinematic aspects of compressional (P) wave propagation within the medium. However, acoustic wave propagation is an oversimplified representation of real-world scenarios, as it solely considers P-wave propagation and does not adequately model the dynamics of the wavefield [8, 9]. Consequently, this oversimplification leads to suboptimal accuracy of the reconstructed medium parameters [10, 11, 12, 13, 14].

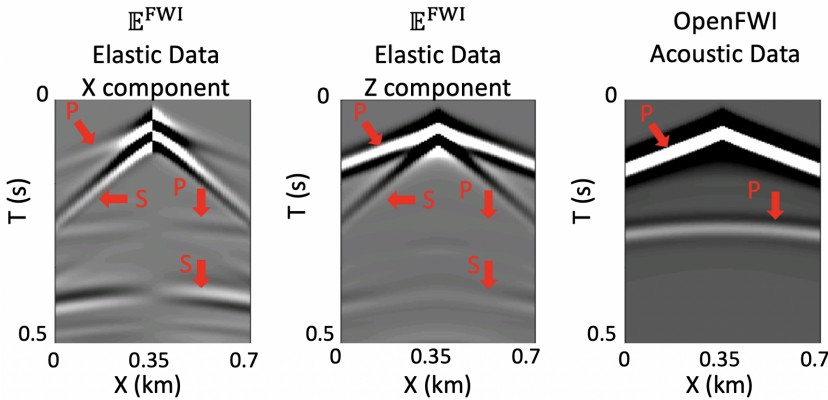

Figure 2: **Comparison of elastic data in $\mathbb{E}^{\mathbf{FWI}}$ and acoustic data in OPENFWI**. Acoustic data only contain P-waves propagation while elastic data contain both P- and S-waves.

**Why elastic FWI:** Elastic inversion, which considers both P- and shear (S-) waves, provides a more comprehensive and precise representation of the subsurface. The correlation between the P-

($V_P$) and S-wave velocities ($V_S$) holds significant implications in the determination of Poisson's ratio (i.e., $V_P$-$V_S$ ratio) and Young's modulus. These parameters play a vital role in the reservoir characterization and serve as essential indicators in the identification and assessment of hydrogen and geothermal reservoirs [15, 16, 17, 18]. The following aspects highlight their significance:

- ***Lithology discrimination***: *Combination of* $V_P$ *and* $V_S$ *velocities is useful for the lithology estimation, while* $V_P$ *alone introduces significant ambiguity because of the overlap of* $V_P$ *for different types of rocks [19].*

- ***Fracture characterization***: *Using the Poisson's ratio (*$V_P$*-*$V_S$ *ratio) and S-wave splitting can estimate fracture orientation and facilitate hydraulic fracturing stimulation [20].*

- ***Estimation of fluid content and saturation***: *Poisson's ratio (*$V_P$*-*$V_S$ *ratio) allows us to estimate the compressibility and estimate the fluid property qualitatively with other relevant reservoir parameters such as the pressure and temperature [21].*

Elastic FWI, as a prominent multiparameter-inversion technique, allows us to simultaneously estimate P- and S-wave velocities [22]. However, the simultaneous consideration of multiple parameters and the expanded dimensions of seismic data significantly increases the complexity of the objective function. This escalation results from the enhanced nonlinearity and the induced trade-offs between the velocities. The coupled impact of P- and S-wave velocities on seismic response further complicates the iterative update process for each parameter. Additionally, the nonlinearity becomes even more pronounced when multiple parameter classes are incorporated into the inversion, as this substantially expands the model space by introducing an increased degree of freedom [23]. Thus, the multidimensionality of elastic FWI renders the problem considerably more complex and challenging compared to the acoustic single-parameter counterpart. With the recent development of machine learning, researchers have been actively exploring *data-driven* solutions for multiparameter FWI, including multilayer perceptron (MLP) [24], encoder-decoder-based convolutional neural networks (CNNs) [25, 26], recurrent network [27, 28], generative adversarial networks (GANs) [29], etc. Nonetheless, the absence of a publicly available elastic dataset poses challenges in facilitating a fair comparison of these methods.

To illustrate the essence of these parameters and the efficacy of elastic FWI, we spotlight the Gallery of $\mathbb{E}^{\mathbf{FWI}}$ in Figure 1. This visualization showcases four sets of samples, each hailing from one of the datasets $\mathbb{E}^{\mathbf{FVB}}$, $\mathbb{E}^{\mathbf{FFB}}$, $\mathbb{E}^{\mathbf{CVB}}$, $\mathbb{E}^{\mathbf{CFB}}$. Each set encompasses three distinct subplots:

- ***P-wave Velocity Map*** *(*$V_P$*)::* *Demonstrating the speed at which Primary or P-waves traverse, these velocities provide insights into the composition and layering of the subsurface, such as the presence of fluids or gas.*

- ***S-wave Velocity Map*** *(*$V_S$*)::* *Reflecting the pace of Secondary or S-waves, these velocities are sensitive to the rigidity and shear strength of the geological formations, offering a more detailed perspective on rock and sediment characteristics.*

- ***Poisson's Ratio Map*** *(*$Pr$*)::* *Derived from the* $V_P$ *and* $V_S$ *maps, this visualization quantifies the subsurface's ability to deform under compressive stress. A higher Poisson's ratio typically indicates a more ductile material, while a lower value suggests a more brittle nature, making this measure crucial for understanding the geomechanical behavior of subsurface formations.*

$\mathbb{E}^{\mathbf{FWI}}$ is constructed upon our previously published open-access acoustic seismic dataset, known as OPENFWI [30]. Our approach incorporates the advantageous characteristics of *multi-scale*, *multi-domain*, and *multi-subsurface-complexity*, inherited from the OPENFWI framework. Furthermore, $\mathbb{E}^{\mathbf{FWI}}$ entails the creation of S-wave velocity maps and employs the elastic wave equation in the forward modeling phase (Figure 2). The computational demands associated with conducting elastic forward modeling are substantial. Consequently, the availability of this dataset would significantly alleviate the burden on researchers.

$\mathbb{E}^{\mathbf{FWI}}$ facilitates equitable comparisons across various methodologies using multiple datasets. In this study, we evaluate the effectiveness of three prominent methodologies derived from pre-existing networks, namely InversionNet [31], VelocityGAN [32], and SimFWI [33]. The objective of this evaluation is to establish a benchmark for future investigations. For comprehensive replication attempts, including the GitHub repository, pre-trained models, and associated licenses, we direct readers to the resources referenced in Section 1 of supplementary materials.

The rest of this paper is organized as follows: Section 2 offers a comprehensive overview of the fundamental principles governing elastic FWI. Section 3 presents a detailed description of the methodology employed in the construction of the dataset. Section 4 offers a succinct introduction to three deep learning methods employed for benchmarking purposes, alongside the presentation of inversion performance on each dataset. The investigation of the interdependence between P- and S-waves is conducted through ablation experiments, as outlined in Section 5. Section 6 outlines the challenges faced and discusses the future implications of the dataset. Lastly, Section 7 offers conclusive remarks summarizing the key findings and contributions.

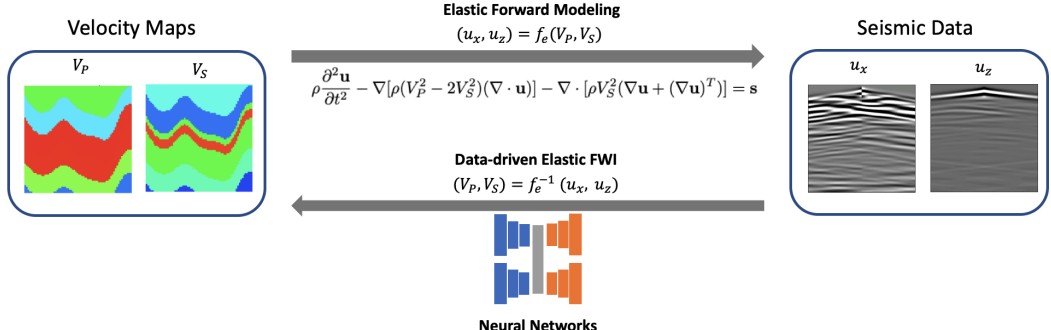

Figure 3: **Schematic depiction of the data-driven approach for elastic forward modeling and FWI**. The forward modeling process involves utilizing elastic forward modeling to compute seismic data by employing the governing elastic wave equations, while elastic FWI employs neural networks to infer the P- and S-wave velocity maps from seismic data containing vertical and horizontal components.

## 2    Elastic Forward Modeling and Data-driven FWI

Figure 3 provides a concise illustration of 2D data-driven elastic FWI and the relationship between P-, S-wave velocity maps and the input horizontal and vertical components of particle displacement therein. In general, the objective of data-driven elastic FWI is to employ neural networks to determine the subsurface velocity maps of the P- ($V_P$) and S-waves ($V_S$). The velocities depict the propagation speed of P- and S-waves through the subsurface medium and depend on the spatial coordinates $(x, z)$. We also consider the density of the subsurface as $\rho$. The source term, represented by **s**, depends on spatial coordinates and time $(x, z, t)$. This source term excites both the P- and S-wave components. The particle displacement in horizontal and vertical directions is denoted by the vector $\mathbf{u} = (u_x, u_z)$. The governing equation for the elastic wave forward modeling in an isotropic medium is given as [34]:

$$\rho \frac{\partial^2 \mathbf{u}}{\partial t^2} - \nabla[\rho(V_P^2 - 2V_S^2)(\nabla \cdot \mathbf{u})] - \nabla \cdot [\rho V_S^2(\nabla \mathbf{u} + (\nabla \mathbf{u})^T)] = \mathbf{s}, \tag{1}$$

In the above equation:

- $\nabla$ represents the gradient calculation of a scalar field in space, specifically:

$$\nabla f = \left[\frac{\partial f}{\partial x}, \frac{\partial f}{\partial z}\right]^T$$

- $\nabla \cdot$ denotes the divergence of a vector field in space, which is expressed as:

$$\nabla \cdot \mathbf{v} = \frac{\partial v_x}{\partial x} + \frac{\partial v_z}{\partial z}$$

For simplicity, we assume a constant density $\rho$ with the value of $1 \ g/cm^3$. The forward modeling problem can be expressed as $(u_x, u_z) = f_e(V_P, V_S)$, where $f_e(\cdot)$ signifies the highly nonlinear elastic forward modeling. It details how the P- and S-waves, initiated by the source $s$, navigate through the subsurface characterized by $V_P$ and $V_S$ over time $t$. Receivers then record these waves as components $u_x$ and $u_z$. The ultimate goal of data-driven elastic FWI is to harness neural networks

Table 1: **Dataset summary of** $\mathbb{E}^{\mathbf{FWI}}$. Velocity maps are represented in dimensions of depth $\times$ width $\times$ length, while seismic data is presented as #sources $\times$ time $\times$ #receivers in width $\times$ #receivers in length.

| Group | Dataset | Size | #Train/#Test | Seismic Data Size | Velocity Map Size |
|---|---|---|---|---|---|
| $\mathbb{E}^{\mathbf{Vel}}$ | $\mathbb{E}^{\mathbf{FVA/B}}$ | 123GB | 24K / 6K | $5 \times 1000 \times 1 \times 70$ | $70 \times 1 \times 70$ |
| Family | $\mathbb{E}^{\mathbf{CVA/B}}$ | 123GB | 24K / 6K | $5 \times 1000 \times 1 \times 70$ | $70 \times 1 \times 70$ |
| $\mathbb{E}^{\mathbf{Fault}}$ | $\mathbb{E}^{\mathbf{FFA/B}}$ | 222GB | 48K / 6K | $5 \times 1000 \times 1 \times 70$ | $70 \times 1 \times 70$ |
| Family | $\mathbb{E}^{\mathbf{CFA/B}}$ | 222GB | 48K / 6K | $5 \times 1000 \times 1 \times 70$ | $70 \times 1 \times 70$ |

to learn the inverse mapping $(V_P, V_S) = f_e^{-1}(u_x, u_z)$. This inverse process enables us to deduce the subsurface velocity maps ($V_P$ and $V_S$) from the recorded particle displacements ($u_x$ and $u_z$) acquired from the receivers. By training neural networks with datasets of recorded waveforms and matching velocity maps, we can fine-tune the network parameters for a precise estimation of the subsurface velocities.

## 3 $\mathbb{E}^{\mathbf{FWI}}$ Dataset

This section describes the methodology used to extend the velocity maps from the OPENFWI dataset to elastic FWI and generate our new dataset $\mathbb{E}^{\mathbf{FWI}}$. Our intention is to provide an accessible, open-source benchmark dataset that can comprehensively facilitate the development and evaluation of the machine learning algorithms in elastic FWI.

The basic information and physical meaning of all the datasets in $\mathbb{E}^{\mathbf{FWI}}$ is summarized in Table 1 and Table 2. The velocity maps encompass P- $V_P$ and S-wave velocities $V_S$, whereas the seismic data comprise the horizontal and vertical components of particle displacement, $u_x$ and $u_z$. The geophysical attributes in "$\mathbb{E}^{\mathbf{Vel}}$ Family" and the "$\mathbb{E}^{\mathbf{Fault}}$ Family" have been constructed utilizing the $V_P$ maps derived from two distinct groups, namely the "*Vel* Family" and the "*Fault* Family" within the OPENFWI dataset. Similar to OPENFWI, the dataset has been categorized into two distinct versions, namely easy (A) and hard (B), based on the relative complexity of the subsurface structures. A thorough examination of the methodologies employed in the construction of the $V_P$ maps and the detailed analysis of the complexity inherent in the velocity maps can be found in [30].

The P-wave velocity ($V_P$) maps in $\mathbb{E}^{\mathbf{FWI}}$ are identical to those in the previously published OPENFWI dataset. For example, $V_P$ in $\mathbb{E}^{\mathbf{FVA}}$ is corresponding to "FlatVel-A" in OPENFWI, $V_P$ in $\mathbb{E}^{\mathbf{CFB}}$ is corresponding to "CurveFault-B" in OPENFWI, and the same naming rule applies to the rest datasets. These velocity maps incorporate a wide range of geological scenarios reflecting diverse subsurface complexities, thereby providing an extensive testbed for machine learning methodologies.

In order to construct the S-wave velocity ($V_S$) maps, we incorporate the Poisson's ratio (Pr) maps [35], which provide a representation of the relationship between the P- ($V_P$) and the S-wave velocities ($V_S$)

$$P_r = \frac{V_P^2 - 2V_S^2}{2V_P^2 - 2V_S^2}.$$

(2)

The initial step involves the generation of Poisson's ratio (Pr) maps by selecting two values within the reasonable range of 0.1 to 0.4 in a random manner [36]. One of these values is allocated to represent the background, whereas the other value is assigned to represent a thin-layer reservoir. Thin-layer reservoirs are selected due to their significance in representing areas where pores are saturated with fluids, making them crucial targets for subsurface exploration and reservoir detection. In the $\mathbb{E}^{\mathbf{FWI}}$ framework, the S-wave velocity ($V_S$) maps are synthesized by multiplying the models of P-wave velocity ($V_P$) with the respective Poisson's ratio (Pr) maps, adhering to the following relationship:

$$V_S = \sqrt{\frac{0.5 - Pr}{1 - Pr}} * V_P.$$

(3)

This approach ensures a wide range of velocity contrasts, resulting in diverse wavefield behaviors, thus expanding the scope of scenarios for machine learning tests in elastic FWI. The details of the elastic forward modeling are given in Section 2 of the supplementary materials.

Table 2: **Physical meaning of $\mathbb{E}^{\mathbf{FWI}}$ dataset**

| Dataset | Grid Spacing | Velocity Map Spatial Size | Source Spacing | Source Line Length | Receiver Line Spacing | Receiver Line Length | Time Spacing | Recorded Time |
|---|---|---|---|---|---|---|---|---|
| $\mathbb{E}^{\mathbf{Vel}}$, $\mathbb{E}^{\mathbf{Fault}}$ Family | $5\ m$ | $0.35 \times 0.35\ km^2$ | $87.5\ m$ | $0.35\ km$ | $5\ m$ | $0.35\ km$ | $0.001\ s$ | $1\ s$ |

## 4 $\mathbb{E}^{\mathbf{FWI}}$ Benchmarks

### 4.1 Deep Learning Methods for Elastic FWI

Our benchmark presents inversion results by three deep learning-based approaches, namely $\mathbb{E}$lasticNet, $\mathbb{E}$lasticGAN, and $\mathbb{E}$lasticTransformer. These methods are derived from pre-existing networks, namely InversionNet [31], VelocityGAN [32], and SimFWI [33], with modifications tailored to address the challenges posed by elastic FWI. We provide a summary of each method separately as follows:

$\mathbb{E}$**lasticNet** is extended from the vanilla InversionNet [31] to the elastic setting with two pairs of input and output. It is a fully-convolutional neural network taking seismic data $u_x$ and $u_z$ as the input of two encoders to learn the latent embeddings independently. The mutual representations of two inputs are concatenated and then forwarded to two independent decoders to obtain the estimated velocity maps $V_P$ and $V_S$ as output.

$\mathbb{E}$**lasticGAN** follows the design of VelocityGAN [32] but substitutes the original generator with an encoder-decoder network such as $\mathbb{E}$lasticNet. The estimated velocity maps $V_P$ and $V_S$ produced by the generator are fed to two independent discriminators to identify the real and fake predictions. A CNN architecture is employed for both discriminators.

$\mathbb{E}$**lasticTransformer** follows a similar seismic-encoder and velocity-decoder architecture design as the SimFWI described in [33]. It consists of two two-layer transformer encoders that take $u_x$ and $u_z$ as inputs and two two-layer transformer decoders to output $V_P$ and $V_S$ separately. Two latent embeddings of $u_x$ and $u_z$ are concatenated and passed through two Maxout converters, then transformed embeddings fed into the decoders. Unlike the linear upsampler utilized at the end of the velocity decoder in [33], we stack upsampling and convolution blocks to construct the upsampler.

### 4.2 Inversion Benchmarks

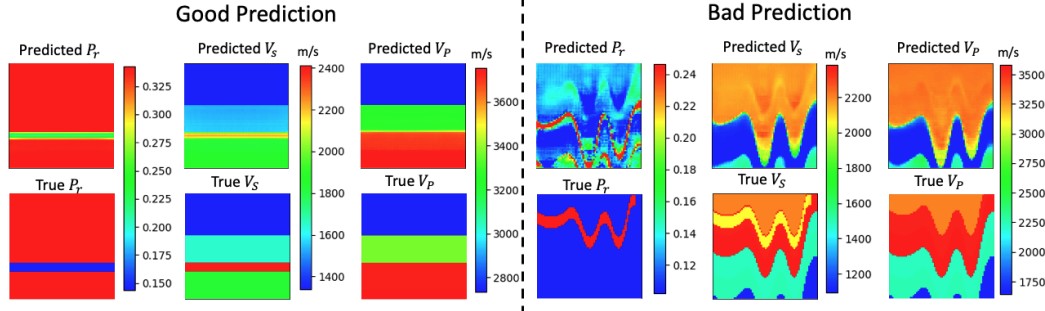

Figure 4: **Examples of both successful and inadequate predictions** in $\mathbb{E}^{FWI}$ benchmarks performed by the $\mathbb{E}$lasticNet.

The experiments were conducted using Nvidia Tesla V100 GPUs, and the training parameters were kept consistent across all datasets. The training process is conducted utilizing the $\ell_1$-norm and $\ell_2$-norm loss functions, respectively. In our study, we assess not only the accuracy of the predicted velocities $V_P$ and $V_S$ but also the degree of decoupling between them by evaluating the accuracy of the predicted Poisson ratio $(\mathrm{Pr})$. To quantify the performance of our predictions, we utilize three evaluation metrics: mean absolute error (MAE), root mean square error (RMSE), and structural similarity index (SSIM). These metrics provide a comprehensive assessment of the quality of our predictions and their similarity to the ground truth values. The performance of $\mathbb{E}$lasticNet on various datasets is presented in Table 3, while Table 4 provides the estimated training time per

epoch for each method on the $\mathbb{E}^{FWI}$ datasets. In Figure 4, examples of inverted velocity maps obtained using the $\mathbb{E}$lasticNet are presented alongside the corresponding ground truth velocity maps. These visual representations highlight instances where the inversion process successfully predicts accurate velocities, as well as instances where further improvement is required. The benchmarks with $\mathbb{E}$lasticGAN and $\mathbb{E}$lasticTransformer are given in Section 6 of the supplementary materials.

The performance of all three models exhibits a decline as the complexity of the dataset increases. Notably, in the case of the dataset with version B, it consistently exhibits lower performance compared to the dataset with version A. The network provides direct predictions for $V_P$ and $V_S$, whereas $Pr$ is obtained indirectly through calculations based on $V_P$ and $V_S$. As a result, $Pr$ consistently exhibits lower SSIM compared to $V_P$ and $V_S$. However, it should be noted that $Pr$ represents a sparser map compared to $V_P$ and $V_S$, leading to lower MAE and RMSE values for $Pr$ compared to $V_P$ and $V_S$.

Table 3: **Quantitative results** of $\mathbb{E}$lasticNet on $\mathbb{E}^{\mathbf{FWI}}$ datasets.

| Dataset | Loss | ElasticNet | | | | | | | | |
|---|---|---|---|---|---|---|---|---|---|---|
| | | $V_P$ | | | $V_S$ | | | $Pr$ | | |
| | | MAE↓ | RMSE↓ | SSIM↑ | MAE↓ | RMSE↓ | SSIM↑ | MAE↓ | RMSE↓ | SSIM↑ |
| $\mathbb{E}^{\mathbf{FVA}}$ | $\ell_1$ | 0.0308 | 0.0559 | 0.9615 | 0.0259 | 0.0500 | 0.9596 | 0.0329 | 0.0664 | 0.8455 |
| | $\ell_2$ | 0.0235 | 0.0455 | **0.9702** | 0.0196 | 0.0385 | **0.9683** | 0.0307 | 0.0583 | **0.8644** |
| $\mathbb{E}^{\mathbf{FVB}}$ | $\ell_1$ | 0.0668 | 0.1468 | **0.8891** | 0.0483 | 0.1053 | **0.8951** | 0.0542 | 0.1057 | **0.7138** |
| | $\ell_2$ | 0.1016 | 0.1901 | 0.8354 | 0.0691 | 0.1322 | 0.8599 | 0.0756 | 0.1302 | 0.6227 |
| $\mathbb{E}^{\mathbf{CVA}}$ | $\ell_1$ | 0.0745 | 0.1345 | 0.8055 | 0.0600 | 0.1080 | 0.8051 | 0.0574 | 0.1156 | 0.5766 |
| | $\ell_2$ | 0.0745 | 0.1343 | **0.8033** | 0.0616 | 0.1087 | **0.8020** | 0.0604 | 0.1131 | **0.6131** |
| $\mathbb{E}^{\mathbf{CVB}}$ | $\ell_1$ | 0.1722 | 0.2982 | 0.6529 | 0.1258 | 0.2165 | 0.6827 | 0.0915 | 0.1580 | **0.4612** |
| | $\ell_2$ | 0.1682 | 0.3048 | **0.6566** | 0.1234 | 0.2220 | **0.6875** | 0.0956 | 0.1660 | 0.4337 |
| $\mathbb{E}^{\mathbf{FFA}}$ | $\ell_1$ | 0.0543 | 0.1026 | **0.9042** | 0.0647 | 0.1349 | 0.8225 | 0.0710 | 0.1501 | **0.6447** |
| | $\ell_2$ | 0.0937 | 0.1537 | 0.8607 | 0.0769 | 0.1309 | **0.8305** | 0.0830 | 0.1369 | 0.6251 |
| $\mathbb{E}^{\mathbf{FFB}}$ | $\ell_1$ | 0.1198 | 0.1859 | 0.7014 | 0.0947 | 0.1462 | 0.7346 | 0.0802 | 0.1312 | 0.4902 |
| | $\ell_2$ | 0.1084 | 0.1704 | **0.7131** | 0.0811 | 0.1290 | **0.7523** | 0.0719 | 0.1225 | **0.5270** |
| $\mathbb{E}^{\mathbf{CFA}}$ | $\ell_1$ | 0.0551 | 0.1128 | **0.8814** | 0.0518 | 0.1042 | **0.8445** | 0.0528 | 0.1150 | **0.6562** |
| | $\ell_2$ | 0.0972 | 0.1636 | 0.8223 | 0.0886 | 0.1390 | 0.7891 | 0.0927 | 0.1443 | 0.5536 |
| $\mathbb{E}^{\mathbf{CFB}}$ | $\ell_1$ | 0.1535 | 0.2307 | 0.5981 | 0.1123 | 0.1698 | 0.6408 | 0.1012 | 0.1602 | 0.3576 |
| | $\ell_2$ | 0.1562 | 0.2305 | **0.6160** | 0.1138 | 0.1697 | **0.6608** | 0.0854 | 0.1393 | **0.4490** |

## 5 Ablation Study

### 5.1 Independent vs. Joint Inversion: Impact on $Pr$ Maps

The first experiment examined the impact of separate versus joint inversion of $V_P$ and $V_S$ on the accuracy of predicted Poisson ratio ($Pr$) maps. This process involved individually training two InversionNets on the $\mathbb{E}^{\mathbf{FWI}}$ dataset to predict $V_P$ and $V_S$ maps, which were then used to calculate the $Pr$ maps. The results revealed a substantial deterioration in map quality, with the independent inversion maps exhibiting significantly higher MAE and MSE, and lower SSIM values, as outlined in Table 5, compared to those reconstructed from joint inversion, shown in Table 3, especially for the complex B datasets, such as "$\mathbb{E}^{\mathbf{CFB}}$". These findings reinforce the significance of considering the $V_P$-$V_S$ relationship and P-S wave coupling, with the single-parameter inversion approach being deemed unviable. Detailed information on this experiment can be found in Section 6 of the supplementary materials.

Table 4: **Training time** in each epoch by each benchmarking method on $\mathbb{E}^{\mathbf{FWI}}$ datasets. All the models are trained on a single GPU.

| | ElasticNet | ElasticGAN | ElasticTransformer |
|---|---|---|---|
| $\mathbb{E}^{\mathbf{Vel}}$ Family | 4m15s | 2m20s | 1m15s |
| $\mathbb{E}^{\mathbf{Fault}}$ Family | 8m35s | 3m50s | 2m30s |

Table 5: **Quantitative results** of OPENFWI's InversionNet trained with $\mathbb{E}^{FWI}$ data. Input z-component data and output the $V_P$/$V_S$ maps independently. The performance is a benchmark of $\mathbb{E}$lasticNet.

| Dataset | Loss | InversionNet | | | | | | | | |
|---|---|---|---|---|---|---|---|---|---|---|
| | | $V_P$ | | | $V_S$ | | | Pr | | |
| | | MAE↓ | RMSE↓ | SSIM↑ | MAE↓ | RMSE↓ | SSIM↑ | MAE↓ | RMSE↓ | SSIM↑ |
| $\mathbb{E}^{\mathbf{FVA}}$ | $\ell_1$ | 0.0392 | 0.0712 | **0.9455** | 0.0239 | 0.0447 | **0.9590** | 0.0461 | 0.0885 | **0.8282** |
| | $\ell_2$ | 0.0451 | 0.0745 | 0.9414 | 0.0251 | 0.0469 | 0.9585 | 0.0541 | 0.1039 | 0.8071 |
| $\mathbb{E}^{\mathbf{FVB}}$ | $\ell_1$ | 0.1030 | 0.1986 | 0.8260 | 0.0643 | 0.1318 | 0.8615 | 0.1290 | 0.2773 | 0.6063 |
| | $\ell_2$ | 0.0883 | 0.1832 | **0.8453** | 0.0620 | 0.1269 | **0.8665** | 0.0905 | 0.1980 | **0.6603** |
| $\mathbb{E}^{\mathbf{CVA}}$ | $\ell_1$ | 0.1016 | 0.1699 | **0.7636** | 0.0736 | 0.1245 | **0.7837** | 0.1050 | 0.2141 | **0.5607** |
| | $\ell_2$ | 0.1052 | 0.1730 | 0.7460 | 0.0720 | 0.1236 | 0.7798 | 0.1194 | 0.2381 | 0.5454 |
| $\mathbb{E}^{\mathbf{CVB}}$ | $\ell_1$ | 0.1854 | 0.3266 | 0.6270 | 0.1388 | 0.2405 | 0.6578 | 0.1523 | 0.3064 | **0.4695** |
| | $\ell_2$ | 0.1820 | 0.3260 | **0.6323** | 0.1331 | 0.2344 | **0.6674** | 0.1553 | 0.3230 | 0.4648 |
| $\mathbb{E}^{\mathbf{FFA}}$ | $\ell_1$ | 0.0818 | 0.1413 | 0.8625 | 0.0584 | 0.1034 | **0.8681** | 0.0784 | 0.1585 | **0.6937** |
| | $\ell_2$ | 0.0788 | 0.1343 | **0.8930** | 0.0946 | 0.1525 | 0.7918 | 0.1351 | 0.2485 | 0.6174 |
| $\mathbb{E}^{\mathbf{FFB}}$ | $\ell_1$ | 0.1323 | 0.2001 | 0.6790 | 0.0943 | 0.1453 | 0.7312 | 0.1180 | 0.2283 | 0.5280 |
| | $\ell_2$ | 0.1301 | 0.1979 | **0.6808** | 0.0898 | 0.1382 | **0.7399** | 0.1124 | 0.2095 | **0.5494** |
| $\mathbb{E}^{\mathbf{CFA}}$ | $\ell_1$ | 0.1012 | 0.1638 | 0.8624 | 0.0710 | 0.1182 | **0.8485** | 0.0778 | 0.1586 | **0.6857** |
| | $\ell_2$ | 0.0962 | 0.1663 | **0.8443** | 0.0833 | 0.1394 | 0.8106 | 0.1032 | 0.1911 | 0.6393 |
| $\mathbb{E}^{\mathbf{CFB}}$ | $\ell_1$ | 0.1702 | 0.2485 | **0.6020** | 0.1243 | 0.1807 | 0.6531 | 0.1317 | 0.2378 | **0.5091** |
| | $\ell_2$ | 0.1745 | 0.2563 | 0.5849 | 0.1219 | 0.1775 | **0.6588** | 0.1379 | 0.2529 | 0.4841 |

## 5.2 Investigating P- and S-waves Coupling via Machine Learning

The second experiment focused on examining the interaction between P- and S-waves in the context of seismic data inversion. Two InversionNets were trained, one focusing on P-wave velocity ($V_P$) and the other on S-wave velocity ($V_S$), while adjusting the structural characteristics of the ignored wave. This experiment, trained with the OPENFWI's InversionNet using data from $\mathbb{E}^{\mathbf{FWI}}$, revealed that any minor change in the disregarded wave velocity structure significantly degraded the network's performance, as evidenced in Table 6. This outcome was clearly demonstrated in the more complex datasets, such as "$\mathbb{E}^{\mathbf{CFB}}$" test set, where changes in structure led to a substantial increase in MAE and RMSE, along with a decrease in the SSIM. For a more detailed analysis, refer to the supplementary materials.

# 6 Discussion

## 6.1 Future Challenge

**Decouple P- and S-waves:** The interaction between P- and S-waves during seismic wave propagation poses a significant challenge when attempting to simultaneously determine P and S velocities. The networks described in this paper exhibit limited success in separating P- and S-waves within the seismic data. Consequently, we anticipate the development of robust methodologies that can precisely estimate both P and S velocities while effectively mitigating the interdependence between these wave components.

**Generalization of data-driven methods:** The elastic approximation provides a more accurate representation of field data in comparison to acoustic data. As a result, we expect the neural networks trained on the $\mathbb{E}^{\mathbf{FWI}}$ dataset to exhibit improved resilience in handling real-world field data. However, it should be noted that there are additional physical phenomena, such as anisotropy and viscosity, which are not accounted for in the $\mathbb{E}^{\mathbf{FWI}}$ dataset. The question of how to incorporate these phenomena into the analysis of field data remains an open and unanswered challenge.

**Forward modeling:** The computational expense associated with elastic forward modeling surpasses that of acoustic cases due to various factors. These include the increased memory requirements and the implementation of smaller grid sizes to counteract dispersion phenomena, among others. A detailed comparison highlighting these aspects can be found in the last section of supplementary

Table 6: **Quantitative results** of InversionNet trained with $\mathbb{E}^{\mathbf{FWI}}$ data. The performance compared between testing on the same and different disregarded velocity structural datasets.

| Dataset | Loss | InversionNet | | | | | |
|---|---|---|---|---|---|---|---|
| | | $V_P$: different $V_S$ structure | | | $V_S$: different $V_P$ structure | | |
| | | MAE↓ | RMSE↓ | SSIM↑ | MAE↓ | RMSE↓ | SSIM↑ |
| $\mathbb{E}^{\mathbf{FVA}}$ | $\ell_1$ | 0.1162 | 0.1919 | **0.8974** | 0.2040 | 0.2716 | **0.7977** |
| | $\ell_2$ | 0.1245 | 0.2101 | 0.8849 | 0.2189 | 0.2928 | 0.7793 |
| $\mathbb{E}^{\mathbf{FVB}}$ | $\ell_1$ | 0.2098 | 0.3527 | 0.7086 | 0.2479 | 0.3285 | **0.7182** |
| | $\ell_2$ | 0.1940 | 0.3265 | **0.7279** | 0.2540 | 0.3382 | 0.7022 |
| $\mathbb{E}^{\mathbf{CVA}}$ | $\ell_1$ | 0.1624 | 0.2590 | **0.7233** | 0.2202 | 0.2924 | 0.6794 |
| | $\ell_2$ | 0.1678 | 0.2647 | 0.7043 | 0.2249 | 0.2987 | **0.6824** |
| $\mathbb{E}^{\mathbf{CVB}}$ | $\ell_1$ | 0.3014 | 0.4828 | **0.5206** | 0.3136 | 0.4246 | **0.5398** |
| | $\ell_2$ | 0.3067 | 0.4900 | 0.5181 | 0.3207 | 0.4349 | 0.5341 |
| $\mathbb{E}^{\mathbf{FFA}}$ | $\ell_1$ | 0.1741 | 0.2732 | 0.7464 | 0.2200 | 0.2865 | **0.7362** |
| | $\ell_2$ | 0.1303 | 0.2052 | **0.8490** | 0.2228 | 0.2913 | 0.6952 |
| $\mathbb{E}^{\mathbf{FFB}}$ | $\ell_1$ | 0.1575 | 0.2291 | **0.6501** | 0.2350 | 0.3030 | 0.6480 |
| | $\ell_2$ | 0.1627 | 0.2336 | 0.6492 | 0.2319 | 0.2982 | **0.6517** |
| $\mathbb{E}^{\mathbf{CFA}}$ | $\ell_1$ | 0.1404 | 0.2244 | 0.7951 | 0.2327 | 0.3059 | **0.7360** |
| | $\ell_2$ | 0.1319 | 0.2134 | **0.8159** | 0.2399 | 0.3194 | 0.7139 |
| $\mathbb{E}^{\mathbf{CFB}}$ | $\ell_1$ | 0.1937 | 0.2798 | **0.5747** | 0.2465 | 0.3215 | 0.5785 |
| | $\ell_2$ | 0.1928 | 0.2788 | 0.5613 | 0.2445 | 0.3183 | **0.5915** |

materials. Despite the possibility of bypassing extensive forward modeling by providing the $\mathbb{E}^{\mathbf{FWI}}$ dataset, there remains a need to explore an efficient forward modeling algorithm to accommodate the growing volume of data in the field.

## 6.2 Broader Impact

**Multiparameter inversion:** Multiparameter inversion techniques have found wide-ranging applications across diverse scientific and engineering domains, including but not limited to geophysics, medical imaging, and material science. The introduction of $\mathbb{E}^{\mathbf{FWI}}$ serves as a catalyst for further investigation and the pursuit of innovative methodologies in these fields. By addressing the inherent limitations and complexities associated with multiparameter inversion, this advancement encourages ongoing research and the exploration of novel solutions.

**Carbon-zero emission:** The attainment of carbon-zero emissions holds paramount significance in addressing climate change, safeguarding human well-being, and fostering sustainable development. While researchers continue to explore effective strategies towards achieving this goal, elastic FWI emerges as a promising approach that can contribute significantly. Particularly, elastic FWI plays a crucial role in assessing and developing geothermal energy resources, as well as in facilitating carbon capture and storage projects, among other applications. The introduction of $\mathbb{E}^{\mathbf{FWI}}$ as a fundamental dataset for elastic FWI is expected to stimulate further research and innovation in this direction, thereby enhancing our understanding and capabilities in the pursuit of carbon-zero emissions.

**New energy exploration:** Elastic FWI can be utilized to evaluate the geological viability of potential sites for hydrogen storage, including underground formations or depleted oil and gas reservoirs. The suitability, capacity, and feasibility of a storage site heavily rely on the effectiveness of a geophysical survey and characterization approaches. With the availability of the $\mathbb{E}^{\mathbf{FWI}}$ dataset, it would yield great potential to enhance the accuracy in characterizing the subsurface reservoir, therefore providing better identification of hydrogen storage locations.

**Potential Social Impacts:** Our research, while advancing inverse problems in natural science, may inadvertently support increased fossil fuel consumption if applied to optimize oil and gas drilling, raising environmental concerns. However, the same techniques can equally bolster positive initiatives, such as geothermal exploration and hydrogen storage, as highlighted in Sec 6.2. The dual potential of

our methodologies underscores the importance of their judicious application, aligning with sustainable development goals.

## 7   Conclusion

This paper presents $\mathbb{E}^{\textbf{FWI}}$, an open-source elastic FWI dataset. $\mathbb{E}^{\textbf{FWI}}$ comprises eight datasets and includes benchmarks for three deep learning methods. The datasets released with $\mathbb{E}^{\textbf{FWI}}$, provide diverse P-wave and S-wave velocities, specifically addressing the coupling problem encountered in multiparameter inversion. The initial benchmarks demonstrate promising results on certain datasets, while others may require further investigation. Additionally, coupling tests are conducted to provide insights into network design for multiparameter inversion problems. Furthermore, this paper discusses the future challenges that can be explored using these datasets and outlines the envisioned future advancements as $\mathbb{E}^{\textbf{FWI}}$ continues to evolve.

## Acknowledgement

This work was funded by the Los Alamos National Laboratory (LANL) - Technology Evaluation and Demonstration (TED) program and by the U.S. Department of Energy (DOE) Office of Fossil Energy's Carbon Storage Research Program via the Science-Informed Machine Learning to Accelerate Real-Time Decision Making for Carbon Storage (SMART-CS) Initiative.

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
