# $\mathbb{E}^{\textbf{FWI}}$: Multi-parameter Benchmark Datasets for Elastic Full Waveform Inversion of Geophysical Properties
## Supplemental Material

**Shihang Feng**[1,*]     **Hanchen Wang**[1,*]     **Chengyuan Deng**[1,2]     **Yinan Feng**[1]

**Yanhua Liu**[1,3]     **Min Zhu**[1,4]     **Peng Jin**[1,5]     **Yinpeng Chen**[6]     **Youzuo Lin**[1,7]

[1]Los Alamos National Laboratory  [2]Rutgers University  [3]Colorado School of Mines
[4]University of Pennsylvania  [5]The Pennsylvania State University  [6]Microsoft
[7]University of North Carolina at Chapel Hill
`shihang.feng@live.com`
`{hanchen.wang, charles.deng, ynf, yanhualiu, minzhu, pjin, ylin}@lanl.gov`
`yiche@microsoft.com, yzlin@unc.edu`

Supplementary materials arrangement:

- Section 1 provides an overview of the public resources available to facilitate reproducibility and outlines the licenses associated with the $\mathbb{E}^{\textbf{FWI}}$ data and released code.

- Section 2 explains the particular method we utilized for elastic seismic forward modeling, which enabled us to generate the seismic data.

- Section 3 provides detailed information on the format and naming conventions used for $\mathbb{E}^{\textbf{FWI}}$.

- Section 4 provides a comprehensive description of the intricate architecture of $\mathbb{E}$lasticNet, $\mathbb{E}$lasticGAN, and $\mathbb{E}$lasticTransformer.

- Section 5 outlines the specific training details employed for $\mathbb{E}$lasticNet, $\mathbb{E}$lasticGAN, and $\mathbb{E}$lasticTransformer.

- Section 6 showcases the benchmark results achieved by $\mathbb{E}$lasticGAN and $\mathbb{E}$lasticTransformer.

- Section 7 explores an ablation study focusing on the impacts of independently inverting $\mathbf{V_P}$ and $\mathbf{V_S}$, thereby sidelining their coupling effects, utilizing the $\mathbb{E}^{\textbf{FWI}}$ datasets.

- Section 8 delves into an ablation study investigating the interdependencies and influence of variations in $\mathbf{V_P}$ and $\mathbf{V_S}$ structures on seismic data.

- Section 9 presents comprehensive computational details regarding elastic forward modeling.

- Section 10 presents a comparison between $\mathbb{E}^{\textbf{FWI}}$ and OPENFWI.

# 1  $\mathbb{E}^{\textbf{FWI}}$ Public Resources and Licenses

To ensure the reliability of reproducing $\mathbb{E}^{\textbf{FWI}}$ benchmarks, we have established several accessible resources. These resources are summarized in the following list. Additionally, our dedicated team is actively engaged in maintaining the platform and incorporating future advancements based on valuable feedback from the community.

---

*Equal contribution

37th Conference on Neural Information Processing Systems (NeurIPS 2023) Track on Datasets and Benchmarks.

- **Website:** `https://efwi-lanl.github.io`
- **Dataset URL:** `https://efwi-lanl.github.io/#dataset`

Codes and pretrained model will be released upon approval by Los Alamos National Laboratory and U.S. Department of Energy.

## 2 Seismic Data Generation

In $\mathbb{E}^{\mathbf{FWI}}$, the seismic source and receiver geometries remain aligned with the OPENFWI dataset [1], except that the grid spacing is reduced to $5m$ in order to preserve the stability of elastic wave propagation. Each velocity map is associated with a total of 5 seismic sources and 70 receivers, which are evenly distributed on the upper surface. This configuration ensures an abundance of source-receiver pairs for the purpose of seismic data generation.

Our Python forward modeling algorthm follows the Matlab code at `https://csim.kaust.edu.sa/files/ErSE328.2013/LAB/Chapter.FD/lab.fdpsv/lab2.html`. The seismic data is simulated from the velocity maps using finite-difference solver [2] with the elastic equations [3] with a 2nd-order accuracy in time and a 4th-order accuracy in space. A 350 grids absorbing boundary [4] is adopted to avoid the reflections from the model boundaries. This method provides a robust and computationally efficient mean of generating accurate seismic data that align with the $V_P$ and $V_S$ models. The point source function utilized in this study is a Ricker wavelet with a central frequency of 15 Hz. This particular wavelet is applied to the vertical component of particle displacement.

## 3 $\mathbb{E}^{\mathbf{FWI}}$ Datasets: Illustration, Format, Naming

The $\mathbb{E}^{\mathbf{FWI}}$ datasets are organized into eight folders, namely FVA, FVB, CVA, CVB, FFA, FFB, CFA, and CFB. These folders contain the datasets $\mathbb{E}^{\mathbf{FVA}}$, $\mathbb{E}^{\mathbf{FVB}}$, $\mathbb{E}^{\mathbf{CVA}}$, $\mathbb{E}^{\mathbf{CVB}}$, $\mathbb{E}^{\mathbf{FFA}}$, $\mathbb{E}^{\mathbf{FFB}}$, $\mathbb{E}^{\mathbf{CFA}}$, and $\mathbb{E}^{\mathbf{CFB}}$, respectively. The examples of $\mathbf{V_P}$, $\mathbf{V_S}$ velocity maps, along with seismic data $u_x$ and $u_z$ in $\mathbb{E}^{\mathbf{FWI}}$ are shown in Figure 1 to 4.

**Format:** All samples in $\mathbb{E}^{\mathbf{FWI}}$ are stored in `.npy` format. The velocity maps, denoted as $V_P$ and $V_S$, as well as the Poisson's ratio $Pr$, and the seismic data $u_x$ and $u_z$, are all stored separately in individual files for preservation. Each file contains a single NumPy array of 500 samples. The shapes of the arrays in velocity map files, Poisson's ratio, and seismic data files are (500, 1, 70, 70) and (500, 5, 1000, 70), respectively.

**Naming:** The naming of files can be described as `{vp|vs|pr|data_x|data_z}_{i}.npy`, where `vp`, `vs`, `pr`, `data_x` and `data_z` specify whether a file stores $V_P$, $V_S$, $Pr$, $u_x$ or $u_z$, `i` is the index of a file (start from 0) among the ones with the same `n`. Here are several examples:

- `vp_3.npy` is the third file among all the files with $V_P$ velocity maps.
- `vs_1.npy` is the first file among all the files with $V_S$ velocity maps.
- `pr_4.npy` is the fourth file among all the files with Poisson's ratio $Pr$ maps.
- `data_x_1.npy` is the file that contains the seismic data x component $u_x$ corresponding to the velocity maps in `vp_1.npy` and `vs_1.npy`.
- `data_z_1.npy` is the file that contains the seismic data z component $u_z$ corresponding to the velocity maps in `vp_1.npy` and `vs_1.npy`.

## 4 $\mathbb{E}^{\mathbf{FWI}}$ Benchmarks: Network Architecture

### 4.1 $\mathbb{E}$lasticNet

$\mathbb{E}$lasticNet is an encoder-decoder structural CNN network built upon InversionNet [5]. The architecture consists of two encoders that take seismic data $u_x$ and $u_z$ as inputs, representing the horizontal and vertical components respectively. The encoder comprises a stack of 14 CNN layers. The first layer has a kernel size of $7 \times 1$, while the subsequent six layers have a kernel size of $3 \times 1$. To reduce the data dimension to the size of the velocity map, a stride of 2 is applied every other layer. Following

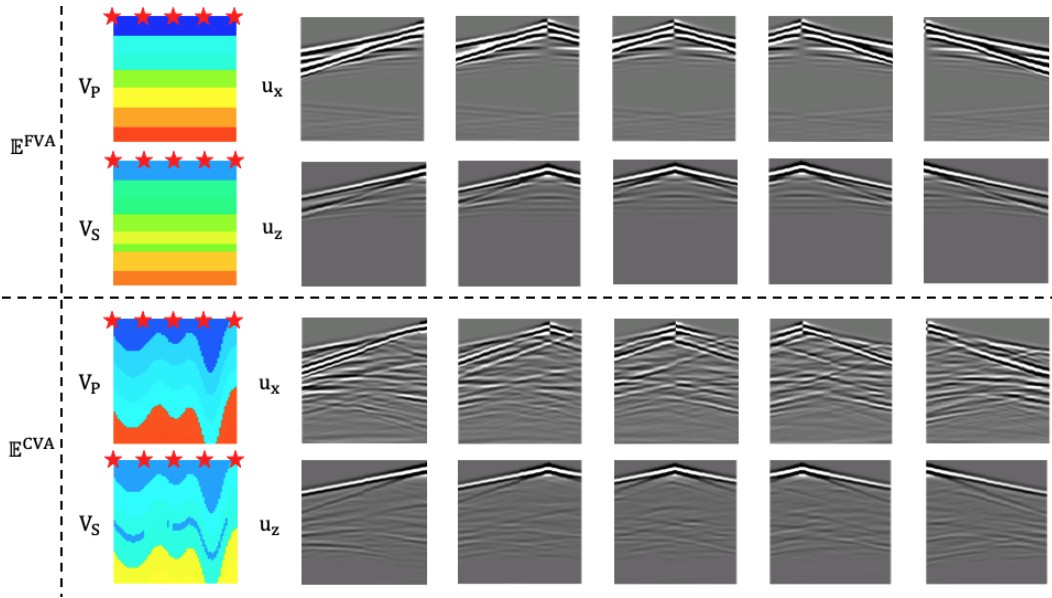

Figure 1: **Examples of $V_P$ and $V_S$ maps, along with seismic data $u_x$ and $u_z$, in $\mathbb{E}^{\mathbf{FVA}}$ and $\mathbb{E}^{\mathbf{CVA}}$.** The star markers indicate the source locations.

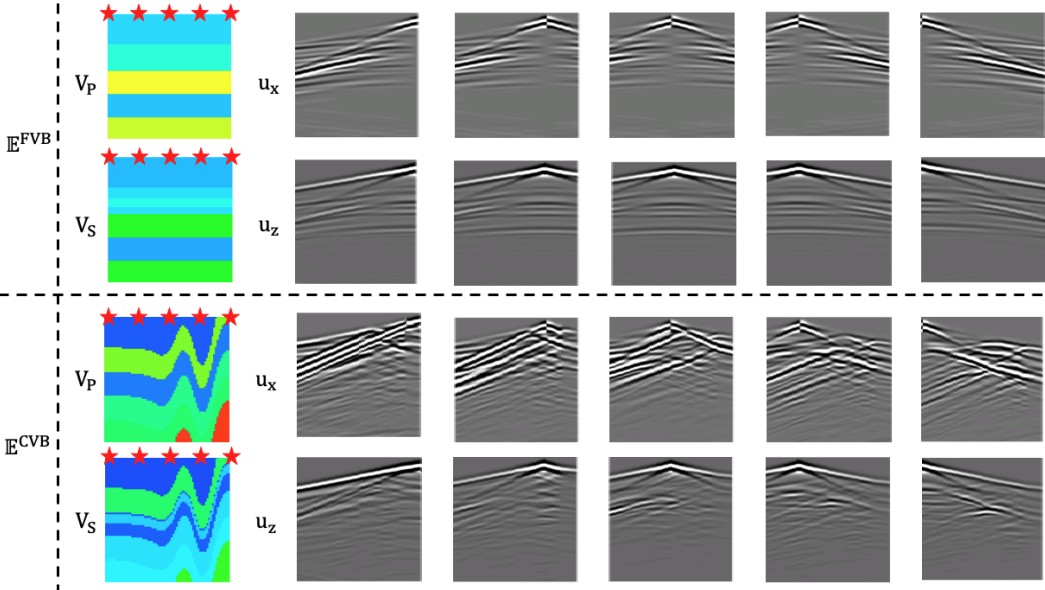

Figure 2: **Examples of $V_P$ and $V_S$ maps, along with seismic data $u_x$ and $u_z$, in $\mathbb{E}^{\mathbf{FVB}}$ and $\mathbb{E}^{\mathbf{CVB}}$.** The star markers indicate the source locations.

this, six additional CNN layers with a kernel size of $3 \times 3$ are employed to extract spatial-temporal features from the compressed data. In these layers, the data is down-sampled every other layer using a stride of 2. Afterward, a CNN layer with an $8 \times 9$ kernel size is used to flatten the feature maps into a latent vector of size 512. The latent vectors from both encoders are concatenated and passed through two decoders to obtain P- and S-wave velocity maps, denoted as $V_P$ and $V_S$ respectively. In the decoder, the latent vector undergoes a deconvolutional layer to generate a $5 \times 5 \times 512$ tensor using a kernel size of 5. This is followed by a convolutional layer with the same number of input and output channels. This deconvolution-convolution process is repeated four times using a kernel size of 4 for the deconvolutional layers. As a result, a feature map of size $80 \times 80 \times 32$ is obtained. Finally, the feature map is center-cropped using a $70 \times 70$ window, and a $3 \times 3$ convolutional layer is applied to generate a single-channel velocity map. The overall architecture consists of 14 CNN layers in the

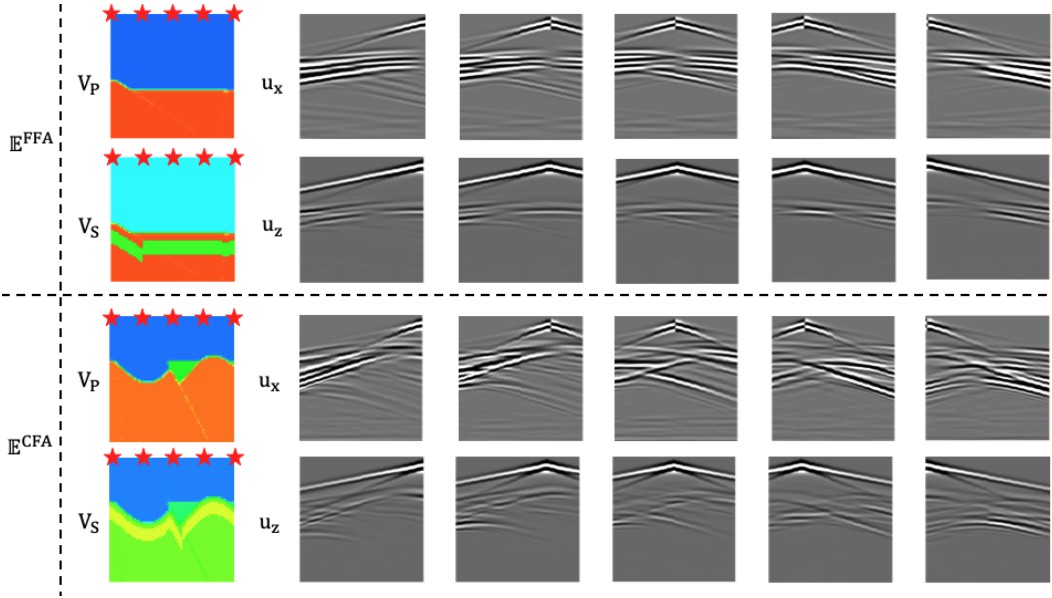

Figure 3: **Examples of** $V_P$ **and** $V_S$ **maps, along with seismic data** $u_x$ **and** $u_z$**, in** $\mathbb{E}^{\mathbf{FFA}}$ **and** $\mathbb{E}^{\mathbf{CFA}}$. The star markers indicate the source locations.

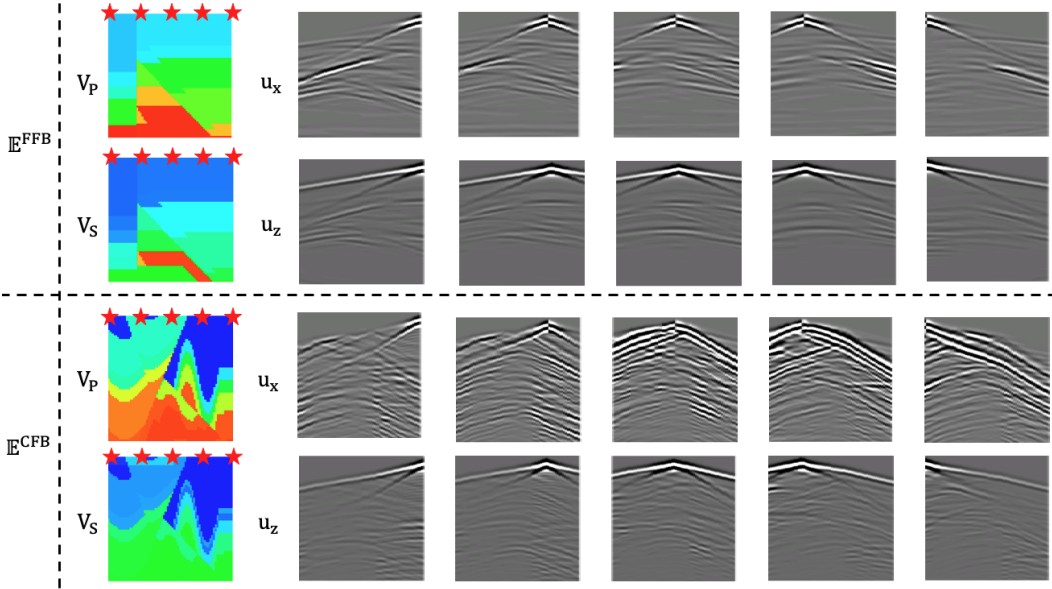

Figure 4: **Examples of** $V_P$ **and** $V_S$ **maps, along with seismic data** $u_x$ **and** $u_z$**, in** $\mathbb{E}^{\mathbf{FFB}}$ **and** $\mathbb{E}^{\mathbf{CFB}}$. The star markers indicate the source locations.

encoder and 11 layers in the decoder. All the convolutional and deconvolutional layers are followed by batch normalization, and the activation function used is leakyReLU.

## 4.2  $\mathbb{E}$lasticGAN

$\mathbb{E}$lasticGAN is extended from the VelocityGAN architecture [6] with an encoder-decoder CNN network as the generator, while the discriminator is a fully-CNN network. Note that the generator allows for other model architecture, though we adopt $\mathbb{E}$lasticNet for the consistency on performance evaluation. The discriminator takes the generated velocity maps ($V_P$, $V_S$) as two inputs and classifies them into fake or true predictions. Each encoder of the discriminator has 9 convolution layers with

LeakyReLU activation but not any normalization. The training process follows the common practice using the Wasserstein distance with gradient penalty, in addition to the pixel-wise $\ell_1$ or $\ell_2$ loss.

### 4.3 $\mathbb{E}$lasticTransformer

$\mathbb{E}$lasticTransformer follows a similar seismic-encoder and velocity-decoder architecture design as the SimFWI described in [7]. It consists of two two-layer transformer encoders that take $u_x$ and $u_z$ as inputs and two two-layer transformer decoders to output $V_P$ and $V_S$ separately. The patch size of seismic data is $100 \times 10$, and the patch size of velocity maps is $10 \times 10$. The dimension of the encoder is 132 with 12 heads, and the dimension of the decoder is 516 with 16 heads. Similarly to SimFWI, we employ a linear layer to project the embedding of $u_x$ and $u_z$ into a 128-dimensional space, separately. These projected embeddings are then concatenated and fed through two separate 2-piece Maxout layers to obtain the latent representations. Subsequently, two additional linear layers are utilized to map each latent representation to the appropriate dimensions of the respective decoders. Unlike the linear upsampler utilized at the end of the velocity decoder in [7], we stack four upsampling and $3 \times 3$ convolution blocks to construct the upsampler. Each block increases the size of the feature map by a factor of two and reduces the dimension by half. In the end there is another $3 \times 3$ convolution layer to generate the single-channel velocity map.

## 5 $\mathbb{E}^{\textbf{FWI}}$ Benchmarks: Training Configuration

This section presents the training configurations that have been implemented to ensure reproducibility. All experiments are conducted using NVIDIA Tesla V100 GPUs. We maintain consistent hyper-parameters across all datasets for $\mathbb{E}$lasticNet, $\mathbb{E}$lasticGAN, and $\mathbb{E}$lasticTransformer. The AdamW optimizer [8] is employed with a weight decay of $1 \times 10^{-4}$ and momentum parameters $\beta_1 = 0.9$ and $\beta_2 = 0.999$ to update all models. During the training process, we apply min-max normalization to rescale the velocity maps and seismic data within the range of $[-1, 1]$. The velocity values for $\mathbf{V_P}$ maps range from $1500\ m/s$ to $4500\ m/s$, while the velocity values for $\mathbf{V_S}$ range from $612\ m/s$ to $3000\ m/s$. The learning rate is set to $1 \times 10^{-3}$ for $\mathbb{E}$lasticNet, $\mathbb{E}$lasticGAN and $\mathbb{E}$lasticTransformer. For $\mathbb{E}$lasticNet and $\mathbb{E}$lasticGAN, there are no weight decay and the batch size is set as 128. For $\mathbb{E}$lasticTransformer, the weight decay is 0.05 and the batch size is 256.

## 6 $\mathbb{E}^{\textbf{FWI}}$ Benchmarks: $\mathbb{E}$lasticGAN and $\mathbb{E}$lasticTransform

The benchmarks of $\mathbb{E}$lasticGAN and $\mathbb{E}$lasticTransform on $\mathbb{E}^{\textbf{FWI}}$ are given in Table 1 and Table 2. $\mathbb{E}$lasticGAN demonstrates superior performance in predicting $V_P$ and $V_S$ compared to $\mathbb{E}$lasticNet, but for $P_r$, the outcomes of $\mathbb{E}$lasticGAN are inferior to $\mathbb{E}$lasticNet. Among the three models, $\mathbb{E}$lasticTransform yields the best results for simple datasets such as $\mathbb{E}^{FVA}$, $\mathbb{E}^{FFA}$, and $\mathbb{E}^{CFA}$. However, as the complexity of velocity maps increases, $\mathbb{E}$lasticTransformer becomes the poorest performing model.

## 7 Independent vs. Joint Inversion: Impact on $P_r$ Maps

This experiment seeks to explore the implications of independent versus joint inversion of $V_P$ and $V_S$ on the precision of predicted $P_r$ maps, thereby underscoring the importance of considering the relationship between $V_P$ and $V_S$, as well as the coupling of P and S waves.

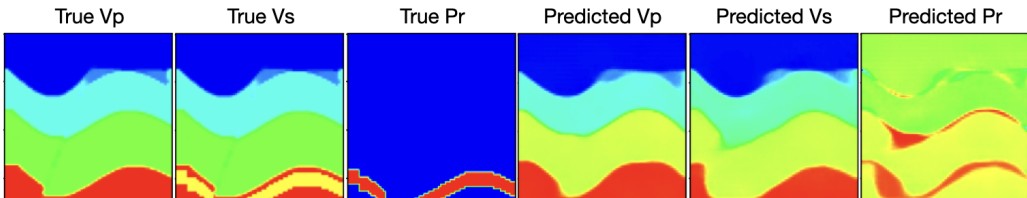

Figure 5: **Examples of independent inversion results of $\mathbb{E}^{\textbf{CFA}}$ dataset:** from left to right: ground truth $V_P$, ground truth $V_S$, ground truth $P_r$, predicted $V_P$, predicted $V_S$, and predicted $P_r$.

Table 1: **Quantitative results** of ElasticGAN on $\mathbb{E}^{\mathbf{FWI}}$ datasets.

| Dataset | Loss | ElasticGAN | | | | | | | | |
| | | Vp | | | Vs | | | Pr | | |
| | | MAE↓ | RMSE↓ | SSIM↑ | MAE↓ | RMSE↓ | SSIM↑ | MAE↓ | RMSE↓ | SSIM↑ |
|---|---|---|---|---|---|---|---|---|---|---|
| $\mathbb{E}^{\mathbf{FVA}}$ | $\ell_1$ | 0.0540 | 0.0882 | **0.9057** | 0.0444 | 0.0809 | 0.8856 | 0.0571 | 0.1129 | **0.6814** |
| | $\ell_2$ | 0.0506 | 0.0774 | 0.8736 | 0.0403 | 0.0623 | **0.9016** | 0.0620 | 0.0960 | 0.5066 |
| $\mathbb{E}^{\mathbf{FVB}}$ | $\ell_1$ | 0.1087 | 0.2012 | **0.8064** | 0.0818 | 0.1520 | **0.8146** | 0.0757 | 0.1329 | **0.5806** |
| | $\ell_2$ | 0.1177 | 0.1975 | 0.7887 | 0.0815 | 0.1458 | 0.8080 | 0.0817 | 0.1289 | 0.4764 |
| $\mathbb{E}^{\mathbf{CVA}}$ | $\ell_1$ | 0.0983 | 0.1594 | **0.7661** | 0.0817 | 0.1331 | 0.7717 | 0.0693 | 0.1258 | **0.5699** |
| | $\ell_2$ | 0.1014 | 0.1524 | 0.7284 | 0.0783 | 0.1204 | **0.7762** | 0.0847 | 0.1309 | 0.4398 |
| $\mathbb{E}^{\mathbf{CVB}}$ | $\ell_1$ | 0.1968 | 0.3293 | **0.6170** | 0.1469 | 0.2427 | **0.6470** | 0.0881 | 0.1505 | **0.4622** |
| | $\ell_2$ | 0.2077 | 0.3218 | 0.6069 | 0.1574 | 0.2386 | 0.6237 | 0.1067 | 0.1563 | 0.3639 |
| $\mathbb{E}^{\mathbf{FFA}}$ | $\ell_1$ | 0.0846 | 0.1452 | 0.8464 | 0.0725 | 0.1204 | 0.8447 | 0.0812 | 0.1394 | 0.5954 |
| | $\ell_2$ | 0.0598 | 0.1015 | **0.8864** | 0.0494 | 0.0833 | **0.8883** | 0.0592 | 0.1017 | **0.6206** |
| $\mathbb{E}^{\mathbf{FFB}}$ | $\ell_1$ | 0.1177 | 0.1781 | **0.6992** | 0.0883 | 0.1373 | **0.7411** | 0.0543 | 0.1016 | **0.6347** |
| | $\ell_2$ | 0.1268 | 0.1798 | 0.6387 | 0.0921 | 0.1338 | 0.7386 | 0.0690 | 0.1102 | 0.4747 |
| $\mathbb{E}^{\mathbf{CFA}}$ | $\ell_1$ | 0.0813 | 0.1471 | 0.8291 | 0.0683 | 0.1234 | 0.8164 | 0.0567 | 0.1179 | **0.6601** |
| | $\ell_2$ | 0.0736 | 0.1176 | **0.8311** | 0.0602 | 0.0989 | **0.8479** | 0.0716 | 0.1148 | 0.5066 |
| $\mathbb{E}^{\mathbf{CFB}}$ | $\ell_1$ | 0.1639 | 0.2366 | **0.6096** | 0.1199 | 0.1741 | **0.6549** | 0.0621 | 0.1055 | 0.5780 |
| | $\ell_2$ | 0.1639 | 0.2326 | 0.6064 | 0.1208 | 0.1726 | **0.6549** | 0.0593 | 0.1009 | **0.6076** |

Table 2: **Quantitative results** of $\mathbb{E}$lasticTransformer on $\mathbb{E}^{\mathbf{FWI}}$ datasets.

| Dataset | Loss | $\mathbb{E}$lasticTransformer | | | | | | | | |
| | | Vp | | | Vs | | | Pr | | |
| | | MAE↓ | RMSE↓ | SSIM↑ | MAE↓ | RMSE↓ | SSIM↑ | MAE↓ | RMSE↓ | SSIM↑ |
|---|---|---|---|---|---|---|---|---|---|---|
| $\mathbb{E}^{\mathbf{FVA}}$ | $\ell_1$ | 0.0326 | 0.0676 | 0.9359 | 0.0232 | 0.0514 | 0.9386 | 0.0351 | 0.0772 | **0.7891** |
| | $\ell_2$ | 0.0337 | 0.0670 | **0.9389** | 0.0240 | 0.0511 | **0.9413** | 0.0367 | 0.0761 | 0.7810 |
| $\mathbb{E}^{\mathbf{FVB}}$ | $\ell_1$ | 0.0830 | 0.1794 | **0.8510** | 0.0595 | 0.1269 | **0.8609** | 0.0692 | 0.1303 | **0.6464** |
| | $\ell_2$ | 0.0871 | 0.1810 | 0.8466 | 0.0641 | 0.1305 | 0.8531 | 0.0723 | 0.1319 | 0.6166 |
| $\mathbb{E}^{\mathbf{CVA}}$ | $\ell_1$ | 0.0826 | 0.1448 | 0.8000 | 0.0648 | 0.1109 | 0.7967 | 0.0959 | 0.1450 | 0.4650 |
| | $\ell_2$ | 0.0853 | 0.1398 | **0.8068** | 0.0659 | 0.1070 | **0.8103** | 0.1034 | 0.1467 | **0.4633** |
| $\mathbb{E}^{\mathbf{CVB}}$ | $\ell_1$ | 0.1772 | 0.3129 | 0.6548 | 0.1294 | 0.2249 | 0.6777 | 0.1225 | 0.1983 | **0.3934** |
| | $\ell_2$ | 0.1838 | 0.2933 | **0.6670** | 0.1354 | 0.2144 | **0.6898** | 0.1363 | 0.1951 | 0.3588 |
| $\mathbb{E}^{\mathbf{FFA}}$ | $\ell_1$ | 0.1190 | 0.1765 | 0.8827 | 0.0779 | 0.1178 | 0.8513 | 0.1544 | 0.1898 | 0.6129 |
| | $\ell_2$ | 0.1153 | 0.1634 | **0.8868** | 0.0671 | 0.1023 | **0.8691** | 0.1418 | 0.1780 | **0.6322** |
| $\mathbb{E}^{\mathbf{FFB}}$ | $\ell_1$ | 0.1120 | 0.1760 | 0.7048 | 0.0802 | 0.1277 | 0.7492 | 0.0960 | 0.1478 | 0.4316 |
| | $\ell_2$ | 0.1161 | 0.1716 | **0.7272** | 0.0819 | 0.1247 | **0.7644** | 0.0886 | 0.1330 | **0.4950** |
| $\mathbb{E}^{\mathbf{CFA}}$ | $\ell_1$ | 0.0372 | 0.0924 | 0.9100 | 0.0365 | 0.0878 | 0.8768 | 0.0422 | 0.1048 | **0.7003** |
| | $\ell_2$ | 0.0601 | 0.1100 | **0.8961** | 0.0498 | 0.0909 | **0.8710** | 0.0621 | 0.1127 | 0.6258 |
| $\mathbb{E}^{\mathbf{CFB}}$ | $\ell_1$ | 0.1863 | 0.2727 | 0.5560 | 0.1388 | 0.2045 | 0.6002 | 0.1433 | 0.2101 | 0.3203 |
| | $\ell_2$ | 0.1787 | 0.2532 | **0.5750** | 0.1302 | 0.1863 | **0.6283** | 0.1284 | 0.1842 | **0.3578** |

The procedure involves the individual training of two separate InversionNets utilizing the $\mathbb{E}^{\mathbf{FWI}}$ dataset. One InversionNet is tasked with the prediction of $V_P$ maps, while the other focuses on the prediction of $V_S$ maps. Subsequently, the independently predicted $V_P$ and $V_S$ outputs are used to compute the Poisson's ratio maps. The derived maps are then statistically compared to the ground truth Pr maps. Further details of these metrics are provided in Table 5 in the main article.

The outcome reveals a significantly higher MAE and MSE, coupled with lower SSIM values for Pr maps reconstructed from independent $V_P/V_S$ predictions when juxtaposed with those reconstructed from joint inversion (Table 3 in the main article). Specifically, the SSIM for the $P_r$ map reconstructed from independent inversion in the "$\mathbb{E}^{\mathbf{FVA}}$" set is 2% less than the $\mathbb{E}$lasticNet joint inversion result in

the $\ell_1$ case and $6\%$ in the $\ell_2$ case. Even in comparison to the InversionNet results in Table 5 in the main article, the Pr performance decrement is still significant. Additionally, both the MAE and MSE of the independent inversion exceed those of the joint inversion result by $60\%$ and by $72\%$ of the independent same structural inversion result. An example visualized comparison is shown in Figure 5. Of critical importance is the observation that key targets of the inversion, the reservoir thin layers, are misplaced at incorrect depths and exhibit distorted spatial shapes in the independent inversion. This misrepresentation could lead to erroneous reservoir exploration, with potentially severe economic ramifications.

Consequently, this experiment reinforces the necessity of incorporating the $V_P$-$V_S$ relationship and the P-S wave coupling in the imaging and targeting of reservoirs. Ignoring the elastic wave coupling and focusing solely on single-parameter inversion is deemed unviable.

# 8 Investigating P- and S-waves Coupling via Machine Learning

The primary objective of this experiment is to elucidate the coupling effects of P and S waves in seismic data. Specifically, we evaluate the feasibility of a $V_P$-only data-driven FWI that overlooks $V_S$ structural alterations. The experimental design is detailed below:

**Training:** We utilize OPENFWI's InversionNet, which is trained using the Z-component seismic data sourced from $\mathbb{E}^{\mathbf{FWI}}$. This setup is identical to the benchmarks set by the OPENFWI's InversionNet, with the exception that our input seismic data incorporates elastic effects. The output is confined to the $V_P$ maps. We utilize $48,000$ and $24,000$ training samples for the "$\mathbb{E}^{Fault}$" and "$\mathbb{E}^{Vel}$" families, respectively, and reserve the remaining samples for testing. Training samples of the "$\mathbb{E}^{CFA}$" set are shown in Figure 6.

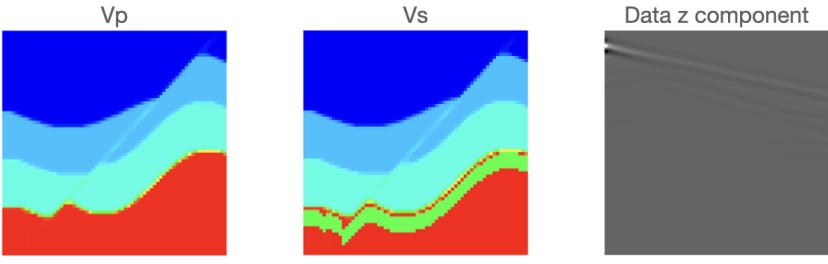

Figure 6: **Examples of independent inversion of $\mathbb{E}^{\mathbf{CFA}}$ training set:** from left to right: ground truth $V_P$, ground truth $V_S$, ground truth seismic data z-component $u_Z$. All used for independent $V_P$, $V_S$ trainings.

**Testing:** The testing phase is divided into two steps: 1) We use the aforementioned reserved test sets as a benchmark to evaluate the performance of InversionNet. 2) We then generate a new elastic seismic dataset by eradicating the thin layer reservoir structure in the Poisson maps, leading to paired $V_P/V_S$ maps with identical structures, shown in Figure 7. This new dataset is identical to the original $\mathbb{E}^{\mathbf{FWI}}$ dataset, save for the minor alterations in $V_S$ structure and the corresponding changes in seismic data. Consequently, the performance differences between the two testing sets should solely reflect the effects of differing $V_S$ structures.

Our observations show a discernible performance decline when testing datasets with differing $V_S$ structures. For instance, in the "$\mathbb{E}^{\mathbf{FVA}}$" set, the new dataset averagely increase by $343\%$ in MAE, and $296\%$ in RMSE compared to the baseline, while SSIM drops by $7\%$ in both $\ell_1$ and $\ell_2$ cases. A detailed account of these statistical metrics is provided in the main article Table 5 and 6.

Conversely, we performed a reciprocal experiment: training on $V_S$-only and testing using the $\mathbb{E}^{\mathbf{FWI}}$ dataset, followed by further testing with altered $V_P$ structures in the thin reservoir layer. The observed outcomes mirror those of the first experiment. With a $V_S$-only InversionNet, when $V_P$ structures are altered slightly in the testing sets, the network's performance notably diminishes. For instance, in the complex case like the "$\mathbb{E}^{\mathbf{CFB}}$" set, the MAE and RMSE for the new dataset increase by $120\%$ and $89\%$ compared to the baseline in the $\ell_1$ case and increase by $115\%$ and $88\%$ in the $\ell_2$ cases, respectively, while SSIM decreases by $7\%$ in both $\ell_1$ and $\ell_2$ cases, respectively. The decrement is even large when compared to the $\mathbb{E}$lasticGAN (Table 1) and the $\mathbb{E}$lasticTransform (Table 2) benchmarks.

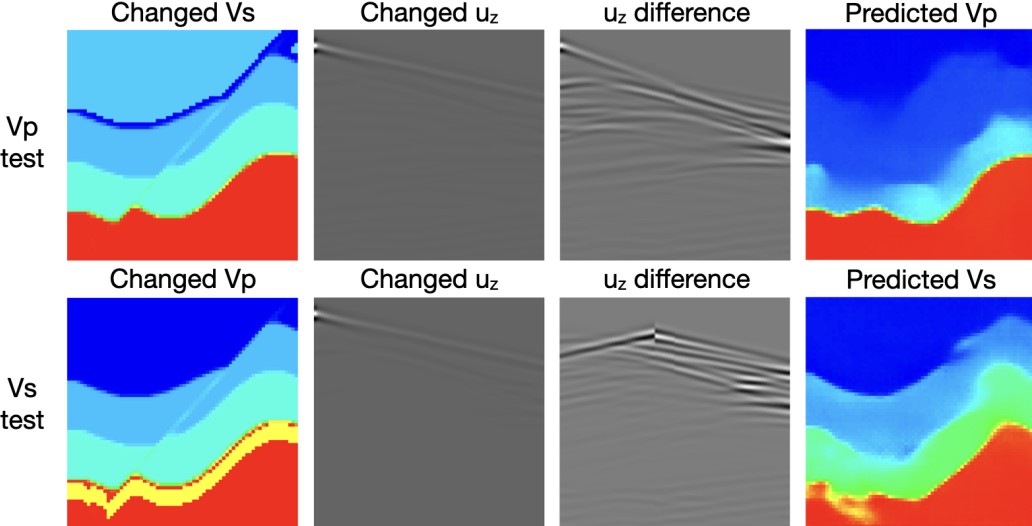

Figure 7: **Examples of independent inversion results of changed structural $\mathbb{E}^{\mathbf{CFA}}$ dataset:** from left to right: changed structural velocities $V_S(V_P)$ for independent $V_P(V_S)$ test, corresponding data z-component $u_Z$ with changed $V_S(V_P)$, corresponding data difference, predicted $V_P(V_S)$ with changed $u_Z$ as input.

It worth mentioning that the diminished Pr prediction performance is based on the same level of Vp and Vs predictions for acoustic and elastic cases. Furthermore, the reservoir layer is not distinctly identifiable in the Pr maps, which could result in significant errors in reservoir estimation, potentially causing substantial economic loss. Detailed statistical metrics are listed in the main article Table 5 and 6.

In summary, single-parameter data-driven inversion, which neglects the coupling of P and S waves, results in substantial degradation of inversion performance. Consequently, the simultaneous prediction of $V_P$ and $V_S$ by considering the coupling of P and S waves proves to be indispensable.

## 9 Computational Cost in Elastic Forward Modeling

The adoption of the elastic approximation reintroduces the concern of computational costs. In the context of data-driven elastic FWI, a significant portion of the computational expenses arises from the construction of the training set. In contrast to elastic forward modeling, which generates particle displacement components $u_x$ and $u_z$ from velocity maps $V_P$ and $V_S$, acoustic forward modeling focuses solely on generating stress $p$ from the P-wave velocity map $V_P$. The acoustic forward modeling process is governed by the acoustic wave equation, which can be expressed as follows:

$$\nabla^2 p - \frac{1}{V_P{}^2} \frac{\partial^2 p}{\partial t^2} = s, \tag{1}$$

where $\nabla^2 = \frac{\partial^2}{\partial x^2} + \frac{\partial^2}{\partial z^2}$, $V_P$ is P-wave velocity map, $p$ is pressure field and $s$ is source term.

When comparing elastic approximation to acoustic approximation, the generation of seismic data imposes substantially higher memory and computational burdens across multiple factors [9].

- **Velocity Maps:** In an acoustic medium, only the P-wave velocity is sufficient to characterize the properties of the medium at a specific location. However, in an elastic medium, two parameters (the P-wave and S-wave velocities), are needed for an accurate description.

- **Seismic data:** Seismic data in the domain of an elastic medium consists of the stress tensor encompassing horizontal and vertical components. Conversely, seismic data within an acoustic scenario primarily encompasses pressure, denoting a scalar quantity. Thus The memory required to store the elastic wavefield is at least twice of the acoustic wavefield.

- **Wave Equation:** The computational burden associated with solving the equation of motion and constitutive equations is notably reduced in acoustic modeling compared to elastic

Table 3: Dataset comparison between $\mathbb{E}^{\mathbf{FWI}}$ and OPENFWI

| Dataset | $\mathbb{E}^{\mathbf{FWI}}$ | OPENFWI |
|---|---|---|
| Dataset Families | $\mathbb{E}^{\mathbf{Vel}}$, $\mathbb{E}^{\mathbf{Fault}}$ | Vel, Fault, Style, Kimberlina |
| Wave Equation | Elastic | Acoustic |
| Total Size | 0.69 TB | 1.83 TB |
| Total Sample | 168 K | 256 K |
| Input | Particle displacement $u_x$ and $u_z$ | Pressure $p$ |
| Output | $V_P$ and $V_S$ | $V_P$ |
| Target | Obtain decoupled $V_P$ and $V_S$ maps | Obtain accurate $V_P$ maps |

modeling. Specifically, the computational cost of elastic modeling is found to be three to six times higher than that of acoustic modeling when both are implemented on an identical grid.

- **Stability:** In order to mitigate numerical dispersion, it is necessary for the grid size to correspond to the minimum velocity within the model, with the minimum Vs for elastic cases, and the minimum Vp for acoustic cases. Consequently, elastic modeling requires a finer grid spacing compared to acoustic modeling. In the context of 2D simulations, the relationship between the number of times the wave equation needs to be solved for acoustic simulations ($N_{acoustic}$) and elastic simulations ($N_{elastic}$) is expressed as follows:

$$\frac{N_{elastic}}{N_{acoustic}} = \left(\frac{V_P^{min}}{V_S^{min}}\right)^3 \qquad (2)$$

The ratio between $\mathbf{V_P}$ and $\mathbf{V_S}$ usually ranges from 1.4 to 2.1 [10, 11, 12], which make $\frac{N_{elastic}}{N_{acoustic}}$ ranges from 2.7 to.9.3.

Elastic wave propagation simulations generally necessitate at least twice the memory compared to acoustic simulations. The computational demands can range from approximately 4.2 to 55.8 times higher. Consequently, the computation and memory requirements for generating $\mathbb{E}^{\mathbf{FWI}}$ are significantly greater than those of OPENFWI.

## 10   $\mathbb{E}^{\mathbf{FWI}}$ vs OPENFWI

While $\mathbb{E}^{\mathbf{FWI}}$ is constructed upon the foundations of OPENFWI, there remain notable distinctions between these two datasets. Beyond the computational expenses associated with the wave equation discussed in Section 9, Table 3 enumerates additional contrasts. Despite the fact that the data size of $\mathbb{E}^{\mathbf{FWI}}$ is smaller than that of OPENFWI, the inherent complexity of the elastic wave equation makes solving for both $\mathbf{V_P}$ and $\mathbf{V_S}$ from particle displacements $u_x$ and $u_z$ significantly more challenging than solving solely for $\mathbf{V_P}$ from pressure $p$.