# OpenReview forum: "$\mathbf{\mathbb{E}^{FWI}}$: Multiparameter Benchmark Datasets for Elastic Full Waveform Inversion of Geophysical Properties"
_NeurIPS.cc/2023/Track/Datasets_and_Benchmarks — NeurIPS 2023 Datasets and Benchmarks Poster_

### Official Review · Reviewer_fegx · 2023-07-22
**Review of Multiparameter Benchmark Datasets for Elastic FullWaveform Inversion of Geophysical Properties**

**Rating:** 7
**Confidence:** 3
**Correctness:** The methodology and presented dataset…
**Clarity:** The paper is well written.

**Strengths:**

The paper is responsive to the conference topic, it is well written, the approach is sound, and overall methodology is appropriate. This submission merits acceptance.

**Additional Feedback:**

Paper is recommended for acceptance.

**Documentation:**

Sufficient description is presented.

**Ethics:**

No ethics concerns are noted.

**Limitations:**

Limitations are adequately addressed.

**Opportunities For Improvement:**

The paper can be accepted in the present form.

**Relation To Prior Work:**

Relation to prior work is described.

**Summary And Contributions:**

This paper presents a benchmark dataset of geophysical properties describing seismic signal propagation. The paper is responsive to the conference topic, it is well written, the approach is sound, and overall methodology is appropriate. This submission merits acceptance.

---

> ### Author Response · Authors · 2023-08-20
>
> We are grateful for the reviewer's compliment on our paper. To make our paper easier to read and complete, we have made a few revisions to our first version manuscript. Please kindly review and give comments.

---

### Official Review · Reviewer_6MVi · 2023-07-22
**Valuable dataset, paper could explain some things more clearly**

**Rating:** 6
**Confidence:** 3
**Correctness:** seems so, but I'm not an expert in th…

**Strengths:**

dataset for an interesting scientific problem, for a range of geological structures along with 3 different deep learning methods as initial benchmarks.

**Additional Feedback:**

none

**Clarity:**

Yes, although as mentioned some more work can be done for readers not familiar with the science.

**Documentation:**

The methods used to transform the OpenFWI data to Elastic  FWI are described, but little information is given about the original data.

**Ethics:**

no ethical problems

**Limitations:**

The comment in checklist 1c) about not having seen any negative impacts about this work is a bit amusing given that I assume a major use of these inverse problems solutions will be in areas like drilling for oil and gas, and extracting these more efficiently, and the world is currently facing a climate crisis because of the use of these products :-). However, positive aspects can be framed as enabling more self-sufficiency in energy, with less dependence on unstable or undemocratic parts of the world, and as pointed out in Sec 6.2 finding geothermal and hydrogen sotrage options.

However, this should not prevent the work from being used, it is just that the authors are maybe not thinking about the impact issues as clearly as they could

**Opportunities For Improvement:**

The paper is generally well-written, but still assumes a lot of knowledge in the domain. Figures 1 presents velocity maps but could do a better job of explaining them to a reader unfamiliar with the subject.
Figure 2 has no units or description on x and y axes.

**Relation To Prior Work:**

There wasn't much discussion about previous datasets. This one seems to be extending the OpenFWI dataset

**Summary And Contributions:**

The paper presents a benchmark dataset for elastic full waveform inversion for a range of geological structures along with 3 different deep learning methods as initial benchmarks

---

> ### Author Response · Authors · 2023-08-20
>
> We appreciate the reviewer's comments and suggestions. We carefully give our response here and revision in the paper accordingly.
>
> Q1. Opportunities For Improvement: We acknowledge the importance of making our paper accessible to a wider audience. In response to your suggestion, we add a similar "sonar" example at the beginning of the "Introduction" section to help readers from other backgrounds understand the seismic full waveform inversion problem in an easier way. We have augmented Figure 1 with more comprehensive explanations to elucidate the velocity maps for those unfamiliar with the domain.
> We have re-plotted Figure 2 with proper units and axes.
>
> Q2. Limitations:
> Thank you for raising this pertinent concern. We wholeheartedly acknowledge the potential implications of the technologies we enable, especially in the context of the ongoing climate crisis. While it's indeed possible for the dataset to be applied to oil and gas exploration tasks, its potential extends far beyond. We designed this dataset to be versatile and specifically suitable for green energy exploration and environmental protection tasks, such as: 1. Carbon Capture, Utilization, and Storage (CCUS); 2. Hydrogen Storage; 3. Geothermal Energy Exploration; 4. Waste Water Burial. While we acknowledge the potential dual-use nature of any technological advancement, our primary intention and vision for $\mathbf{\mathbb{E}^{FWI}}$ lean decidedly towards sustainable and environmentally-conscious applications. We'll ensure that future iterations of our work emphasize these aspects even more explicitly to guide its optimal use. In order to elucidate both the favorable and adverse effects of our work on the environment, we have revised checklist 1c to encompass all conceivable applications of our research outcomes. We have added more discussion of potential social concerns in the "Discussion" section.
>
> Q3. Clarity: Thanks again for the suggestion. As stated in the previous questions, we have added more explanations, like the "sonar" example, to make the work easier for readers from other domains of research.
>
> Q4. Relation To Prior Work \& Documentation: We have discussed the transformation from the previous "OpenFWI" dataset to our dataset in "Section $\mathbf{3~~\mathbb{E}^{FWI}}$ Dataset". We have added one more explanation section of the connection between $\mathbf{\mathbb{E}^{FWI}}$ and "OpenFWI". Please find it in the supplementary material Section 10 \& Table 3.

---

> > ### Comment · Reviewer_6MVi · 2023-08-29
> > **no change to checklist?**
> >
> > Thanks for the improvements. I couldn't see any change to checklist 1c?

---

> > > ### Author Response · Authors · 2023-08-29
> > >
> > > Thank you for your response. We have updated the latest version of main article accordingly. It has a discussion on the potential impact of using the dataset for oil and gas. Please kindly review it.

---

### Official Review · Reviewer_yFAK · 2023-07-29

**Rating:** 7
**Confidence:** 3

**Strengths:**

* **[Useful domain]** Predicting velocity maps from seismic data has multiple geological applications, including earthquake prediction and information for extracting oil and gas.

* **[Gathering collection of open-access data into one dataset]** the paper collects open-access data into their format. Therefore, a researcher interested in this problem can attempt fewer datasets yet having access to more data.

* **[Open-sourcing data that is otherwise computationally-expensive to compute]** the paper explains the formulas for computing the S velocity maps (Vs; equations 2 and 3) -- it seems to be a recurrence relation. Computing the Vs maps is computationally-expensive. Open-sourcing these values (1) standardizes the dataset and (2) allows researchers with lower computational budget to train models.

* **[Baseline Models]** Paper attempts 3 baseline models (based on InversionNet; VelocityGAN, and SimFWI). Further, they mention that they open-source pre-trained models.

**Additional Feedback:**

1. Can you better explain Eq.1 and/or give some background/prelim section ?

2. Is it possible to list more data examples, under some categorization that makes sense to the domain? Easy VS hard seems somewhat arbitrary.

**Clarity:**

Yes, but the clarity can be improved by adding visualizations of datapoints, adding categorizations, and better-explaining Eq.1 to audience coming from other ML subdomains.

**Correctness:**

The paper proposes a dataset (that gathers multiple datasets and adds some prediction target, Vs) and implements 3 baselines. The method looks correct to me. However, some things can be improved (e.g., conducting several runs, and providing complete information to non-domain-experts)

**Documentation:**

The dataset URL is present: https://efwi-lanl.github.io/#dataset -- however, I dont see any accessible code.

**Ethics:**

No concerns

**Limitations:**

Yes, they are discussed in section 6.1.

**Opportunities For Improvement:**

* **[Incomplete Information to non-experts]** this paper has taught me so much about seismic data. I am a seasoned NeurIPS author and reviewer. Many NeurIPS folks, are like myself: math & ML savvy, but without much expertise to some domains e.g. geoscience. I think the paper is lacking some basic background section. E.g., Eq1 is hard to decipher. Partial derivative is used ($\frac{\partial^2}{\partial t^2}$) and at the same time the gradient operator $\nabla$ is used. Why use both? Howcome the $\nabla$ is sometimes suffixed with $\cdot$ only sometimes? What are we taking the $\nabla$ gradient with respect to? also time? In that case, why not stick to the partial derivative notation $\frac{\partial}{\partial t}$?

* **[Lacking categorization statistics]** It would be nice to see some categorization -- paper mentions breaking-up data into "easy" and "hard" -- how is this done? Perhaps, it is better to have some quantitative measures for breaking the data, e.g., curvature measure of the ground-truth Pr?

* **[Visualizations]** The paper should have more visualizations of data (only a couple are there).

* **[Statistical Significance]**  Table 3 shows many cells. However, I'd expect to see some statistical significance numbers (or at least some text around the number of repetitions).

**Relation To Prior Work:**

This work (1) collects multiple data sources [that come from previous contributions], (2) computes new target variables (Vs) and (3) runs SOTA baseline models.

**Summary And Contributions:**

Paper presents a dataset E-FWI, which combines a number of open-source datasets on predicting seismic properties. In addition to combining the open-source datasets, the paper also computes the S-wave velocity map, which is computationally-expensive, and they release the dataset to the community. The paper also develops three baselines on this dataset. Baselines receive as inputs the Seismic Data (particle displacement vectors) and output the velocity maps (Vs and Vp).

---

> ### Author Response · Authors · 2023-08-20
>
> We appreciate the reviewer's comments and suggestions. We carefully give our response here and revision in the paper accordingly.
>
> Q1. Incomplete Information to non-experts: The gradient operator, $\nabla$, computes the rate of change of a scalar field as one moves in space. In equation form, for a scalar field $f(\mathbf{x})$, it is given by: $\nabla f = \left[\frac{\partial f}{\partial x},\frac{\partial f}{\partial y},\frac{\partial f}{\partial z}\right]^T$. This returns a vector field from a scalar field input.
>     The divergence operator, $\nabla \cdot$, measures how much a vector field is diverging or converging at a point. For a vector field $\mathbf{v(x)}$, it is expressed as: $\nabla \cdot \mathbf{v} = \frac{\partial v_x}{\partial x}+\frac{\partial v_y}{\partial y}+\frac{\partial v_z}{\partial z}$. This gives a scalar field from a vector field input.
> Furthermore, when considering physical phenomena, the divergence may signify the presence or absence of sources. For instance, a non-zero divergence at a point indicates a source or sink at that location, while a zero divergence implies a source-free region.When it comes to the notation $\frac{\partial ^2}{\partial t^2}$, it's reserved for denoting the second partial derivative with respect to time. This distinction helps avoid confusion with spatial partial derivative operators like gradients.
>
> Q2. Lacking categorization statistics: We claim "B" sets are "hard" and "A" sets are "easy" because the structures in A vmap samples are simpler, i.e. fewer layers, faults, curves. Thus, the corresponding seismic data is simpler. In the "OpenFWI" paper that was accepted by NeurIPS 2022, the authors have analyzed the velocity map complexity using three metrics: spatial information, gradient sparsity index, and Shannon entropy in their supplementary material, which suggests that the $B-$ datasets are more complex than the paired $A-$ datasets. Thus, we claim for the same categorization. We have the discussion of the categorization in the main paper Section 3.
>
> Q3. Visualizations: Figures will take too much space in the main paper due to the length limit, so we put the full visualizations of the data in the supplementary material Section 3, having the full representation of the $V_P$, $V_S$, $P_r$, $u_x$, and $u_z$, respectively.
>
> Q4. Statistical Significance: We use Table 3 to show one detailed benchmark experiment results of ElasticNet with bold font numbers highlighting the best SSIM performance of each variable map of each dataset, regardless of loss function used. The result table is discussed in Section 4.2. It is designed this way to align to the "OpenFWI" dataset paper \cite{deng2022openfwi} result tables, for example, Table 4.
>
> Q5. Documentation: The code is still under reviewing by Los Alamos National Laboratory and DOE. We will release them once it get approved.

---

> > ### Comment · Reviewer_yFAK · 2023-08-21
> >
> > Thank you for the nice explanations.
> >
> > Please add the writing above (or some version of it) to a prelim or background section.
> >
> > I am raising my score to "Accept.
> >
> > All the best!

---

> > > ### Author Response · Authors · 2023-08-21
> > >
> > > Thank you for raising the score. Your valuable comments really help us a lot. We will add the explanation accordingly to the manuscript.

---

### Author Response · Authors · 2023-08-20

We genuinely appreciate all three reviewers' feedback. We are glad that our manuscript is recognized with the following merits:
- Domain Relevance: Velocity map prediction from seismic data is crucial for various geological applications, including earthquake forecasting and oil/gas extraction insights.
- Unified Dataset Collection: The study aggregates open-access data into a unified format, allowing researchers to tap into a richer dataset without needing to source from multiple disparate sets.
- Computational Accessibility: The authors provide computationally-intensive S velocity maps, enabling standardization of datasets and facilitating researchers with limited computational resources to develop models.
- Robust Baselines: The paper introduces and tests three baseline models, including InversionNet, VelocityGAN, and SimFWI, and additionally open-sources pre-trained models.
- Scientific Contribution: Offers a dataset addressing a pivotal scientific challenge, complemented by a variety of geological structures and three initial deep learning benchmarks.
- Paper Quality: The submission aligns well with the conference theme, boasts clear writing, adopts a rigorous approach, and is methodologically sound, making it worthy of acceptance.

As suggested by the reviewers, we recognize the importance of ensuring our paper is understandable, even to those outside the geoscience domain. Thus, we add a similar ``sonar'' example at the beginning of the "Introduction" section to help readers from other backgrounds understand the seismic full waveform inversion problem in an easier way. Besides, we have the following revision to our original manuscript:
- Clarified Mathematical Concepts: Added detailed explanations about mathematical operators like gradient ($\nabla$) and divergence ($\nabla \cdot$) for non-experts. Distinguished between spatial partial derivative operators and time derivatives.
- Dataset Categorization: Provided rationale behind the categorization of "B" sets as "hard" and "A" sets as "easy" based on geological complexity. Mentioned a previous paper's analysis supporting this categorization.
- Visualizations: Addressed concerns about space constraints by relocating comprehensive visualizations to supplementary material.
- Statistical Significance: Presented benchmark results in Table 3 with the best SSIM performance highlighted. The table's design is aligned with previous related work.
- Code Release: Noted that the code is still under review by institutions and will be released post-approval.
- Accessibility: To make the paper more comprehensible for diverse readers, introduced a "sonar" example at the start and enhanced explanations in figures.
- Addressed Potential Impacts: Accepted and acknowledged potential dual-use implications of the technology, particularly in light of the climate crisis. Emphasized the dataset's design for green energy and environmental tasks and revised relevant sections to underscore sustainable applications.
- Enhanced Clarity: To aid readers from other research domains, additional explanations, including the aforementioned "sonar" analogy, were incorporated.
- Connection with Previous Work: Detailed the evolution from the previous "OpenFWI" dataset to the current one and added supplementary sections to highlight the relationship between the datasets.

---

### Decision · Program_Chairs · 2023-09-22

**Decision:**

Accept (Poster)

**Comment:**

This paper presents a novel benchmark dataset for elastic full waveform inversion (a key technique for Seismic Imaging) and a few  deep learning methods as initial benchmarks. The reviewers agree that this is a very good paper in an interesting domain where not a lot of data is available- which makes it very valuable.
The authors did a good job in addressing all the reviewer questions.